
# Drought evolution characteristics of the Qinghai-Tibet Plateau over the last 100 years based on SPEI

Shengzhen Wang[1], Fenggui Liu[1,2] , Qiang Zhou[1] , Qiong Chen[1], Baicheng Niu[1] , Xingsheng Xia[1]

[1] College of Geographical Sciences, Qinghai Normal University, Xining, 811600, China
[2] Academy of Plateau Science and Sustainability, Xining, 811600, China

*Correspondence to*: Fenggui Liu (lfg_918@163.com)

**Abstract:** The standardized precipitation evapotranspiration index (SPEI) of the Qinghai-Tibetan Plateau was calculated using the CRU4.03 gridded dataset from 1901 to 2018 in this paper. Then, based on the SPEI data, drought on the Qinghai-Tibet Plateau was studied in terms of its spatial and temporal distributions and its changing characteristics over the last 100 years. The results revealed that the precipitation in the southeastern part of the Qinghai-Tibet Plateau has been steadily rising over the last 100 years, in conjunction with only minor temperature shifts. In the northwestern part of the plateau, precipitation has decreased significantly, accompanied by a significant increase in temperature. The drought on the Qinghai-Tibetan Plateau showed a clear gradual increase in aridity from southeast to northwest over the last hundred years. The SPEI also showed distinct seasonal patterns, steadily increasing in spring and summer and decreasing significantly in autumn and winter. In
addition, each season had its own spatial characteristics. The northeastern part of the plateau, except the Qaidam Basin, showed a significant aridity trend in all seasons. A wet trend prevailed in the southeastern and southern areas. Drought on the Qinghai-Tibetan Plateau exhibits apparent cyclical oscillations with a main period of 54 years and has different cyclical characteristics in different seasons.

Keywords: Qinghai-Tibet Plateau; SPEI; Centennial Scale; Drought evolution; Periodic Introduction

## 1    Introduction

Drought is a recurring natural phenomenon (Wilhite et al., 2000) and includes meteorological, agricultural, hydrological, and socioeconomic droughts, with meteorological drought acting as the driver of other forms of drought (Zargar et al., 2011). It can be difficult to determine the beginnings and ends of droughts, and the impacts of droughts can extend over larger geographical areas than those of other natural hazards (Wisser et al., 2009; Brown et al., 2016), making droughts one of the
most costly natural disasters in the world (Keyantash J. et al., 2002). The availability of water to humankind has been reduced due to drought (Wisser et al., 2009; Döll P et al., 2009; Montaseri et al., 2017). Food production is also threatened as a result of reduced irrigation water and available water in the soil (Mishra et al., 2010). In addition, drought can also have certain impacts on ecosystems (Zhang et al., 2012b; Zhang et al., 2012a), resulting in an increase in drought-driven tree mortality around the world (Allen C D et al., 2015) and expected ecosystem transformations from one state to another, such as from





forest to shrubland (Jiang et al., 2013). A number of studies indicate that in the context of future climate change, global droughts

will become more severe and frequent (Dai, 2013; Mu et al., 2014; Zhao et al., 2018), especially in arid and semi-arid regions

(Mishra et al., 2010). As a result, studying the spatial and temporal distribution characteristics of drought in semi-arid and arid

regions is critical.

Many drought indices have been developed and used to quantify drought and track its evolution (Palmer, 1968; Wells et al.,

2004; Mishra et al., 2010; Barua et al., 2011; Sousa et al., 2011). Since its introduction in 2010, the standardized precipitation

evapotranspiration index (SPEI) based on precipitation and temperature (Li et al., 2017b) has been widely used in drought

monitoring in semi-arid and arid areas under climate change in quantitative drought studies (Vicente-Serrano et al., 2013)

owing to its twin advantages of multiple timescales and the ability to measure the effect of temperature on drought at the same

time (Vicente-Serrano et al., 2010).

The Qinghai-Tibet Plateau is situated in Asia's and Europe's hinterlands. This region receives a weak wet signal due to the

influence and limitations of atmospheric circulation and altitude (Fan et al., 2003), as water vapor from the ocean is difficult

to transport to this area (Chen et al., 2011). Thus, this area has a typical continental climate, with most regions receiving less

than 200 mm of annual precipitation (Qian et al., 2011). It is one of the world's most drought-prone areas and is highly

susceptible to global climate change (Zhang et al., 2004). Therefore, we calculated the SPEI of the Qinghai-Tibet Plateau from

1901-2018 based on the CRU4.03 gridded dataset. Then, based on these SPEI data, drought on the Qinghai-Tibet Plateau was

studied in terms of its spatial and temporal distributions and its changing characteristics over the last 100 years. The aim is to

provide decision support and references for rational planning and allocation of water resources and to cope with the drought

risks brought about by future climate change.

## 2   Method

### 2.1 Calculation of the SPEI

SPEI data represent multiscale drought data that combine precipitation and temperature and are calculated using the water

balance equation based on the interpolation between precipitation and potential evapotranspiration.

The calculation steps are as follows.

1)   Potential evapotranspiration (PET) was calculated using the FAO Penman-Monteith method, as follows:

$$PET = \frac{0.408\Delta\ (R_n - G)\ + \gamma \frac{900}{T_{mean} + 273} u_2 (e_s - e_a)}{\Delta + \gamma(1 + 0.34u_2)} \tag{0.0}$$

where PET is the potential evapotranspiration (mm•d⁻¹), $R_n$ is the net radiation of the landmark (MJ•m⁻¹•d⁻¹), G is the soil heat

flux (MJ•m⁻²•d⁻¹), $T_{mean}$ is the daily average air temperature (°C), $u_2$ is wind speed at two meters (m•s⁻¹), $e_s$ is the saturated





water pressure (kPa), $e_a$ is the actual water pressure (kPa), $\Delta$ is the slope of the saturated water pressure curve (kPa•°C), and $\Gamma$

is the dry-wet metre constant (kPa•°C).

2)  Interpolation between monthly precipitation and potential evapotranspiration

$$D_i = P_i - PET_i \qquad (0.0)$$

where Pi is the monthly precipitation, PET is the monthly potential evapotranspiration, and Di is the interpolation.

The data series of Di was normalized to calculate the SPEI corresponding to each value. A log-logistic probability distribution

with three parameters was used because of the possibility of negative values in the original data series Di. The cumulative

function of the log-logistic probability distribution is as follows:

$$F(x) = \left[ 1 + \left( \frac{\alpha}{x - \gamma} \right)^{\beta} \right]^{-1} \qquad (0.0)$$

where the parameters (a, b and c) were fitted using the method of linear moments as follows:

$$\alpha = \frac{(W_0 - W_1)\beta}{\Gamma(1 + 1/\beta)\Gamma(1 - 1/\beta)} \qquad (0.0)$$

$$\beta = \frac{(2W_1 - W_0)}{(6W_1 - W_0 - 6W_2)} \qquad (0.0)$$

$$\gamma = W_0 - \alpha\Gamma(1 + 1/\beta)\Gamma(1 - 1/\beta) \qquad (0.0)$$

In the above expression, W1 is the probability-weighted moment of the original data series Di. The calculation is done by the

following method:

$$W_s = \frac{1}{N} \sum_{i=1}^{N} (1 - F_i)^s D_i \qquad (0.0)$$

$$F_i = \frac{i - 0.35}{N} \qquad (0.0)$$

In the above equation, N is the number of months involved in the calculation.

3)  The cumulative probability density is then normalized to the calculation:

$$P = 1 - F(x) \qquad (0.0)$$

4)  A standard normal distribution of the cumulative probability density function is performed to obtain the SPEI time change

sequence:

$$SPEI = W - \frac{c_0 + c_1 W + c_2 W^2}{1 + d_1 W + d_2 W^2 + d_3 W^3} \qquad (0.0)$$

where

$$\begin{cases} W = \sqrt{-2\ln(P)} & P \leqslant 0.5 \\ W = \sqrt{-2\ln(1 - P)} & P > 0.5 \end{cases} \qquad (0.0)$$





The other constant terms in the formula are defined as $c_0$=2.515517, $c_1$=0.802853, $c_2$=0.010328, $d_1$=1.432788, $d_2$=0.189269,
and $d_3$=0.001308.

## 2.2 Mann-Kendall test

The Mann-Kendall (M-K) test is a non-parametric statistical measure. A non-parametric test, also known as a distribution-free test, has the advantage that it does not require the sample to follow a certain distribution, is not troubled by a few outliers, is more suitable for the type and order variables, and is easy to calculate. It is commonly used to identify sudden shifts and
patterns in a sequence. The calculation steps are as follows:

1)   The order column $s_k$ of a sequential time series and $UF_k$ are calculated. For a time series x with a sample size of n, an order column is constructed as follows:

$$s_k = \sum_{i=1}^{k} r_i , \qquad k = 2, 3, \cdots, n \tag{0.0}$$

where

$$r_i = \begin{cases} +1, & x_i > x_j \\ 0, & x_i \leqslant x_j \end{cases} \quad j = 1, 2, \cdots, i \tag{0.0}$$

It can be seen that the order column $s_k$ is the cumulative count of the numbers whose values at moment $i$ are greater than the values at moment $j$.

The statistics are defined under the assumption of random independence of the time series:

$$UF_k = \frac{[s_k - E(s_k)]}{\sqrt{var(s_k)}}, \qquad k = 1, 2, \cdots, n \tag{0.0}$$

where $UR_k$=0, $E(s_k)$, $var(s_k)$ are the mean and variance in the cumulative $s_k$, a sequence of statistics calculated at $x_1$, $x_2$, ..., $x_n$. Given a significance level $\alpha$, the normal distribution table is checked; if $\mid UF_i \mid > U_a$, there is a significant trend change in the sequence.

2) The time series x is changed into the inverse order $x_n$, $x_{n-1}$ ..., $x_1$ by time series x, and then the calculation of the order series $s_k$ and its standard normal distribution $UB_k$ is repeated, while making $UB_k$ =-$UF_k$ (k=n,n-1,…,1), UB1=0.

3) Given a significance level of $\alpha$ =0.05, then the critical value is u0.05 =±1.96.

## 2.3 Morlet wavelet analyses

Wavelet analyses can be used to analyse time series containing non-smooth powers at many different frequencies (Christopher Torrence et al., 1998). Such analysis can effectively capture the cyclical characteristics of hydrometeorological series at different scales and qualitatively assess the trend of the series some time into the future (Daubechies I., 1990). Wavelet analysis
unfolds a one-dimensional signal in time and frequency, analyses its time-frequency structure, and extracts useful information, preserving the benefits of Fourier analysis while compensating for its flaws (Wei, 2007). The calculation process is as follows:

1)  Wavelet function $\Psi(t) \in L^2(R)$ satisfies the following:



$$UF_k = \frac{[S_k - E(S_k)]}{\sqrt{Var(S_k)}}, \quad k = 1, 2, \cdots, n \tag{0.0}$$

$$\int_{-\infty}^{+\infty} \psi(T) \, dt = 0 \tag{0.0}$$

*Ψ(t)* is the wavelet basis function, and the effect is to find the sub-wavelet by telescoping the scale and time translation:

$$\psi_{a,b}(t) = |a|^{-0.5} \Psi\left(\frac{t-b}{a}\right) \tag{0.0}$$

*Ψ$_{a,b}$ (t)* is the wavelet, *a* is the period length of the wavelet, and *b* is the translation factor in time.

     2)   The sequence in the study is mostly discrete data. Hence, we define the function f(kΔt) (k=1,2...,N), where a is the time interval. The discrete sequence wavelet transform is as follows:

$$UF_k = \frac{[S_k - E(S_k)]}{\sqrt{Var(S_k)}}, \quad k = 1, 2, \cdots, n \tag{0.0}$$

*W$_f$(a,b)* is the wavelet transform coefficient.

## 3     Data and preprocessing

### 3.1 Data sources

**The climate data:**

This paper usesd precipitation, temperature, and potential evapotranspiration data from the Climate Research Unit 4.03 grid data dataset (CRU 4.03), developed by the University of East Anglia, UK (https://crudata.uea.ac.uk/). Grid data have better spatial representativeness and continuity than data from traditional station observations, especially on the Qinghai-Tibet Plateau, where meteorological observation stations are sparse and unevenly distributed. In addition, the quality control and homogeneity controls in the CRU dataset are excellent (Mitchell et al., 2005). Although drought studies and SPEI calculations

need at least 30 years of data, the CRU dataset includes over 100 years of data that perfectly meet this requirement (Li et al., 2017a).

**The drought hazard data:**

Drought hazard data extracted from the "China Meteorological Disasters Dictionary", the "China Meteorological Disasters Yearbook", the climate bulletin of the meteorological bureau, the "China Agricultural Information Network"

(http://www.agri.cn/) and disaster data collected from some counties during the second scientific expedition to the Qinghai-Tibetan Plateau resulted in drought hazard data for the Qinghai-Tibetan Plateau for the period 1950 to 2018.

### 3.2 Data preprocessing

The CRU data were preprocessed with a bilinear interpolation algorithm, which first determined the greyscale values (or red-blue-green (RGB) values) of the nearest four points around the point to be measured and then used two-dimensional linear





interpolation to obtain the values of the point to be measured (Liu et al., 2009). The algorithm processes data with a high
       degree of similarity to actual measured data, according to Wang's analysis (Wang et al., 2017).

## 4    Result analysis

### 4.1 Analysis of the applicability of the SPEI on the Tibetan Plateau

The drought index's regional applicability is a critical requirement for accurately describing regional drought. Drought hazard
event data collected were used to test the applicability of the SPEI in the Qinghai-Tibetan Plateau drought study (Fig. 1).
       The results show that the drought frequency matches the SPEI data quite well. Three periods of severe drought were chosen
       as representative of past years with extreme drought disasters for spatial matching tests to further examine its applicability
       (Fig. 2). The results show that the SPEI data and drought hazard have a good spatial match. From 1977 to 1985, a total of 682
       droughts occurred in 682 counties, mainly in Gansu, Qinghai, and Yunnan, among which Tibet experienced severe drought in
1981-1983. From 1994 to 1999, a total of 791 counties were affected by drought, mainly in Qinghai. Yunnan, and Tibet; from
       2008 to 2009, a total of 210 counties were affected by drought. The conclusions drawn from the spatial and temporal
       characteristics are consistent with the drought years recorded in the disaster literature, and there is a strong spatial match. As
       a result, we assume that data from the SPEI can be used to study drought evolution on the Qinghai-Tibet Plateau.

### 4.2 Characteristics of inter-annual variation in SPEI on the Tibetan Plateau

155   The SPEI was calculated on the scale of the Qinghai-Tibetan Plateau from 1901 to 2018, used temperature, precipitation, and
       potential evapotranspiration data from the CRU dataset, and a 118-year series of SPEI changes was developed (Fig. 3) to
       analyse the evolving characteristics of drought on the Qinghai-Tibetan Plateau from 1901 to the present. The results of the
       SPEI data analysis show a clear spatial heterogeneity in drought on the Tibetan Plateau, with a gradual increase in aridity from
       southeast to northwest over the last hundred years.
160   The characteristics of the SPEI changes over the last 100 years show a constant trend of high and low fluctuations but a
       decreasing trend over the whole timescale, with a decrease of approximately 0.011/10 a. The SPEI shifts around the Qinghai-
       Tibetan Plateau appeared to be flat and constant from 1901 to 1934, with no major variations. However, in three time spans,
       1934 to 1946, 1954 to 1972, and 2005 to 2016, there are consistent downward trends, with the largest decrease of 0.18/10 a
       from 2005 to 2016 and the largest increase of 0.19/10 a from 1946 to 1954. Between 1972 and 2005, there were minor
variations, with an average upward trend of 0.11/10 a (Table 1). Mutation analysis of the SPEI using the M-K test showed that
       the SPEI of the Qinghai-Tibetan Plateau showed a change in 1921 and a significant downward trend after 1938 (above the
       significance level 0.05 threshold) (figure omitted).



## 4.3 Intra-annual variation characteristics of the SPEI on the Tibetan Plateau

The seasons in this article are divided as follows: spring is from March to May, summer is from June to August, autumn is
from September to November, and winter is from December to February, according to meteorological standards.

The SPEI seasonal index trend shows a slow increase of 0.005/10 a in spring on the Qinghai-Tibet Plateau, according to statistical analysis (Fig. 4), with nonsignificant fluctuations between 1901 and 1931. Between 1931 and 1944 and between 1952 and 1972, there were significant decreasing trends, and between 1944 and 1952 and between 1972 and 2005, there were significant increasing trends. There were weak fluctuations in the summer between 1901 and 1938, but they were
nonsignificant, while three time periods – 1938 to 1948, 1954 to 1965 and 2008 to 2015 – showed more significant decreasing trends. However, there were weak upward trends with nonsignificant changes from 1948 to 1954 and from 1965 to 2008. There were similarly weak fluctuations in the autumn from 1901 to 1932, but they were nonsignificant in magnitude. Significant decreasing trends were observed in five time periods: 1932 to 1946, 1951 to 1964, 1972 to 1976, 1985 to 1993 and 2010 to 2017; significant increasing trends were observed in three periods: 1946 to 1951, 1964 to 1972 and 1993 to 2010. There were
significant decreasing trends in winter in four time periods: 1936 to 1949, 1959 to 1971, 1983 to 1987 and 1998 to 2018; there were significant increasing trends in two periods: 1949 to 1959 and 1971 to 1983 (Table 2). The largest decrease was 1.8/10 a between 1983 and 1987 in winter, and the largest increase was 0.77/10 a between 1946 and 1951 in autumn. In general, between 1901 and 1935, the SPEI fluctuated nonsignificantly, but after 1935, it showed a more pronounced pattern, with the greatest decreases occurring in autumn and winter.

The SPEI on the Qinghai-Tibetan Plateau exhibits different spatial evolutionary characteristics in different seasons. The SPEI
decreases substantially in spring from the Aljinshan to the northern part of the Qaidam Basin and from the Ali Plateau to the western part of the northern Tibetan Plateau, with a maximum decrease of 0.10/10 a in the central region situated south of the Aljinshan to the northern part of the Qaidam Basin. On the Songpan Plateau, however, there is a significant increasing trend, with a maximum increase of 0.09/10 a. The SPEI in the western part of the Gangetic Mountains, the western part of the
southern Tibetan valley, and the Yarlung Tsangpo River valley shows a strong tendency to decrease in summer, with a decrease of 0.05/10 a, while the SPEI in the northeastern part of the Qaidam Basin and the eastern part of the southern Tibetan valley significantly increases, with an increase of 0.09/10 a. The SPEI shows a strong tendency to decrease in autumn, with a maximum decrease of 0.11/10 a, from the Ali Plateau to the western part of the northern Qinghai-Tibetan Plateau and the southwestern part of the Alpine Mountain, while the SPEI tends to increase, with a maximum increase of 0.04/10 a, in most
of the central and eastern parts of the Tibetan Plateau. The SPEI shows a significant decrease in winter from the western Ali Plateau to the northern Tibetan Plateau, the central Tibetan valley, and the southern Hengduan Mountains, with a maximum decrease of 0.07/10 a, and exhibits a rising trend with a maximum increase of 0.07/10 a on the northeastern edge of the Qaidam Basin and the Songpan Plateau (Fig. 5).

In general, the seasonal variation in the SPEI on the Qinghai-Tibetan Plateau also has obvious regional variation. The
northeastern part of the Tibetan Plateau, especially the Aljinshan to the northern part of the Qaidam Basin, and the western



part of the Ali Plateau to the northern Qinghai-Tibetan Plateau show a significant trend of increasing drought in all four seasons. The Songpan Plateau region, located in the southwestern part of the plateau, shows a significant wetting trend in all three seasons, except for the summer, when it shows a significant increase in drought. The southern part of the Hengduan Mountains and the Yarlung Tsangpo River valley, located in the south of the plateau, show a significant wetting trend in the three seasons

other than winter, when there is a significant trend towards increased drought. In contrast, the SPEI in the Qaidam Basin region, which is located in the northern part of the plateau, increased significantly in all four seasons, showing a significant wetting trend, which is similar to the findings of Jin (Jin Liya et al., 2004) and Dai (Dai et al., 2013)

**4.4 Periodic analyses of SPEI on the Tibetan Plateau**

This paper uses Morlet wavelet analysis to analyse the SPEI of the Qinghai-Tibetan Plateau on a regular basis to determine

whether there are any periodic variations. According to the results of the three-dimensional plot of SPEI wavelet transform coefficients (Fig. 6) and the wavelet variogram (Fig. 7), there is a clear feature of multi-timescale variation in the Tibetan Plateau's SPEI over the past 100 years, with a main cycle of 54 years over the whole timescale and a secondary main cycle of 17 years. This conclusion passed the white Gaussian noise test at the 0.05 significance level.

**5    Discussion**

**5.1 Characteristics of temperature and precipitation variability on the Tibetan Plateau**

The characteristics of temperature and precipitation changes on the Qinghai-Tibetan Plateau from 1901 to 2018 were examined using data from the CRU4.03 dataset. The trend in the anomaly of average precipitation on the Qinghai-Tibetan Plateau is shown in Figure 8; this trend is not characterised by significant changes in precipitation over approximately the last hundred years. The M-K test is given a significance level of 0.05, i.e., u0.05 = ±1.96. The M-K test was used to test for changes in the

average precipitation anomaly on the Tibetan Plateau. The findings show that during the period 1901-2018, the average precipitation anomaly increased, then decreased, and then increased again. There was a significant increasing trend during 1948-1969 (above the significance level threshold of 0.05), a decreasing trend during 1989-2004, and an increasing trend again after 2004. There were 5 changes in 1910, 1921, 1995, 2012 and 2015 (Fig. 8). As shown, there was no obvious rule governing precipitation over the Qinghai-Tibet Plateau, and positive and negative anomalies alternate, reflecting the complexity of

precipitation shifts on the Qinghai-Tibet Plateau.

In comparison to precipitation, the temperature of the Qinghai-Tibet Plateau has shown a significant upward trend over the last 100 years. The M-K test results on the temperature over the Qinghai-Tibet Plateau show that the temperature of the Qinghai-Tibet Plateau has been rising since 1901 (beyond the significance level threshold of 0.05). After 1928-1968 and 1997, the level of significance even exceeded 0.001 (U0.001 =2.56), and there were three changes in 1935, 1950, and 1994.

Precipitation over most areas of the Qinghai-Tibet Plateau is increasing gradually, and the range of precipitation increases gradually from west to east. The maximum increase in precipitation is 0.016 mm/10 a, and the maximum decrease is 0.08





mm/10 a. The temperature of the whole Qinghai-Tibet Plateau presents an increasing trend, and the temperature change rate gradually decreases from west to east, contrary to the precipitation trend. The Qinghai-Tibet Plateau region does not show an obvious positive trend, the western and central regions show a significant increase in temperature, with the centre of this increase located in the northern Har Goolun Range. In addition, the temperature change in the Qaidam basin area is obvious, which is consistent with the conclusion drawn by Zhang and colleagues (Zhang Wangxiong et al., 2019) (Fig. 9).

The precipitation over the Songpan Plateau and other areas in the southern part of the Qinghai-Tibet Plateau has increased significantly during the past century, but the temperature has not increased significantly. In contrast, in the northern region of the Qinghai-Tibet Plateau, especially the Ali Plateau and the western region of the northern Tibetan Plateau, the northern region of the Kunlun Mountains and the western region of the southern Tibetan Valley, the climate has changed from cold and dry to warm and dry.

## 5.2 Changing characteristics of the SPEI on the Qinghai-Tibet Plateau

In the first half century, the SPEI showed a decreasing trend (0.042/10 a), and in the second half century, it showed a significantly increasing trend (0.039/10 a). However, over the whole timescale, it showed a decreasing trend of approximately 0.011/10 a, indicating that the Qinghai-Tibet Plateau as a whole showed a drying trend in the long time series, which is inconsistent with previous research conclusions (Wu et al., 2005; Zheng et al., 2015; Duan et al., 2016; Dang et al., 2019).

Statistical analysis of the SPEI by season is intended to discuss the annual variation characteristics of the SPEI. The results showed that the SPEI of the Qinghai-Tibet Plateau exhibited a slow increasing trend in spring and summer, with an increasing rate of 0.002-0.005/10 a. There were significant decreasing trends in autumn and winter, with decreasing rates of 0.026/10 a and 0.022/10 a, respectively. All four seasons showed periodic fluctuations in the SPEI. The SPEI data showed that the Qinghai-Tibet Plateau became significantly wetter in the autumn from 1946-1951, and the rate of increase in the SPEI was as high as 0.77/10 a. However, winters from 1959-1971 were significantly drier, and the rate of decrease was 0.53/10 a. In the northeastern part of the Qinghai-Tibet Plateau, drought increased significantly in all four seasons, and the maximum decrease in SPEI was 0.11/10 a in autumn, indicating that the most severe drought trends were those extending from the Altun Mountains to the northern Qaidam Basin and from the Ngari Plateau to the western part of the northern Tibetan Plateau. The Songpan Plateau region, located in the southwestern part of the plateau, showed a significant wetting trend in the three seasons other than summer, when it showed a significant increase in drought. The southern part of the Hengduan Mountains and the Yarlung Tsangpo River valley, located in the south of the plateau, show a significant trend of wetting in the three seasons other than winter, when there was a significant trend towards increased drought. In contrast, the SPEI in the Qaidam Basin region, which is located in the northern part of the plateau, increased significantly in all four seasons, showing a significant wetting trend, similar to the findings of Jin (Jin Liya et al., 2004) and Dai (Dai et al., 2013).

In addition, there is a clear feature of multi-timescale variation in the Tibetan Plateau's SPEI over the past 100 years, with a main cycle of 54 years on the whole timescale and a secondary main cycle of 17 years. This conclusion passed the white Gaussian noise test at the 0.05 significance level.





## 6    Conclusion and discussion

Using the CRU4.03 grid point dataset, this paper discussed the Tibetan Plateau's temperature and precipitation variation characteristics from 1901 to 2018. The findings show that the Tibetan Plateau's precipitation variance from 1901 to 2018 is not especially notable, but the temperature, on the other hand, has exhibited a strong upward trend. For example, the rate of change in precipitation increased gradually from west to east across the plateau, while the rate of change in temperature decreased gradually from west to east, indicating a pattern opposite that of precipitation. The Songpan Plateau region, located in the southeastern part of the Qinghai-Tibet Plateau, has seen a marked increase in precipitation over the last hundred years, but temperature changes have not been very pronounced. However, most of the northwestern part of the plateau, such as the Ali region, the northern part of the Kunlun Mountains and the western part of the southern Tibetan valley, has experienced a significant decrease in precipitation over the last 100 years, accompanied by significant warming. This indicates a gradual change from a cold and dry climate to a warm and humid climate in the northwestern part of the plateau. Hence, a gradual change from a cold and dry climate to a warm and dry climate has occurred in the northwestern part of the plateau.

The SPEI was then calculated for the entire Qinghai-Tibetan Plateau using temperature and precipitation data from the CRU4.03 dataset, and the SPEI was used to examine the characteristics of the Tibetan Plateau's drought evolution from 1901 to 2018. The SPEI was then calculated for the entire Qinghai-Tibetan Plateau using temperature and precipitation data from the CRU4.03 dataset, and the SPEI was used to characterize the evolution of drought on the Qinghai-Tibetan Plateau from 1901 to 2018. On the whole timescale, the SPEI of the Qinghai-Tibetan Plateau fluctuates up and down significantly, but the overall trend is negative, with a decrease of approximately 0.011/10 a. The results of the M-K test show that the SPEI changed abruptly in 1921 and showed a significant decline after 1938. There is clear spatial heterogeneity in drought on the Tibetan Plateau, with a gradual increase in aridity from southeast to northwest over the last hundred years. In addition, the seasonal variation in the SPEI on the Qinghai-Tibetan Plateau also has obvious regional variation. The northeastern part of the Tibetan Plateau, especially the Aljinshan to the northern part of the Qaidam Basin and the western part of the Ali Plateau to the northern Tibetan Plateau, showed a significant trend of increasing drought in all four seasons. The Songpan Plateau region, located in the southwestern part of the plateau, showed a significant wetting trend in the three seasons other than summer, when it showed a significant increase in drought. The southern part of the Hengduan Mountains and the Yarlung Tsangpo River valley, located in the south of the plateau, showed a significant trend of wetting in the three seasons other than winter, when there was a significant trend towards increased drought. In contrast, the SPEI in the Qaidam Basin region, which is located in the northern part of the plateau, increased significantly in all four seasons, indicating a significant wetting trend. This paper uses Morlet wavelet analysis to analyse the SPEI of the Qinghai-Tibetan Plateau on a regular basis to determine whether there are any periodic variations. There is a clear feature of multi-timescale variation in the Tibetan Plateau's SPEI over the past 100 years, with a main cycle of 54 years over the whole timescale and a secondary main cycle of 17 years. This conclusion passed the white Gaussian noise test at the 0.05 significance level.



This paper uses the CRU grid point dataset to calculate the SPEI index applicable to drought analysis of the Qinghai-Tibetan Plateau and analyses the evolutionary characteristics of drought on the Qinghai-Tibetan Plateau over the past hundred years in the context of climate change through accurate reanalysis of the data. It has obvious advantages over the indices calculated from station data and compensates for the inaccurate interpolation caused by the sparse and uneven distribution of the stations on the Tibetan Plateau.

This paper used the CRU grid point dataset to calculated the SPEI applicable to drought analysis of the Tibetan Plateau. This paper also analyses the drought evolution characteristics of the Qinghai-Tibetan Plateau over the past 100 years in the context of climate change through precise reanalysis of the data. It has obvious advantages over the indices calculated from station data and compensates for the problem of inaccurate interpolation due to the sparse and uneven distribution of stations on the Tibetan Plateau. Further comprehensive and systematic analyses will be conducted in the future to assess the ability of different levels of SPEI to indicate drought hazard events, with the aim of revealing the drought hazard thresholds indicated by the SPEI indices and analysing the effects of different factors on drought hazards.

**Code and data availability:** All codes to process the data (R code) and the results themselves are available upon request from the corresponding author.

**Author contribution:** Shengzhen Wang and Fenggui Liu designed the study in consultation with Qiang Zhou. Qiong Chen provided the meteorological data, which Baicheng Niu employed to carry out the analyses and wrote much of the analysis of it. Shengzhen Wang wrote the manuscript with support from Baicheng Niu and Xingsheng Xia.

**Competing interests**: The authors declare that they have no conflict of interest.

**Financial support**: This work was supported by the National Key Research and Development Program of China (NKPs) (grant number 2019YFA0606902); The Second Qinghai-Tibetan Plateau Scientific Expedition and Research (STEP) Program (grant number 2019QZKK0906); The "Strategic Priority Research Program" of the Chinese Academy of Sciences (grant number XDA20040200).





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

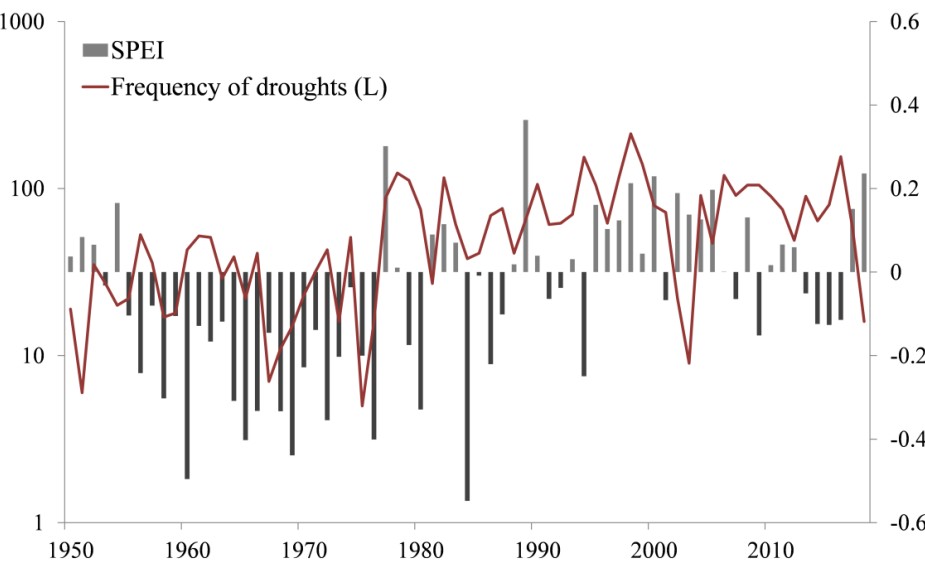

**Fig. 1 SPEI and typical annual drought frequency on the Qinghai-Tibet Plateau**

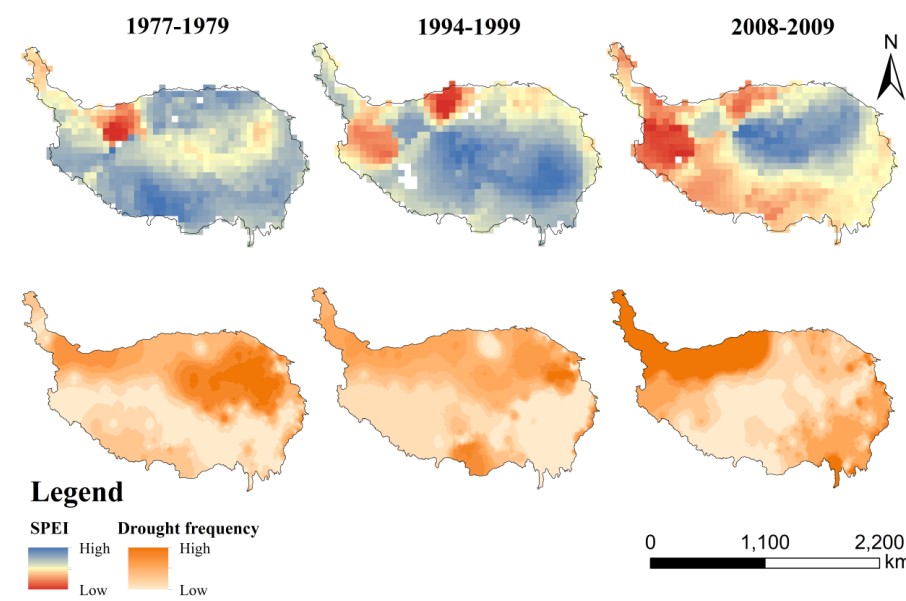

**Fig. 2 SPEI and drought frequency of severe drought years on the Qinghai-Tibet Plateau**

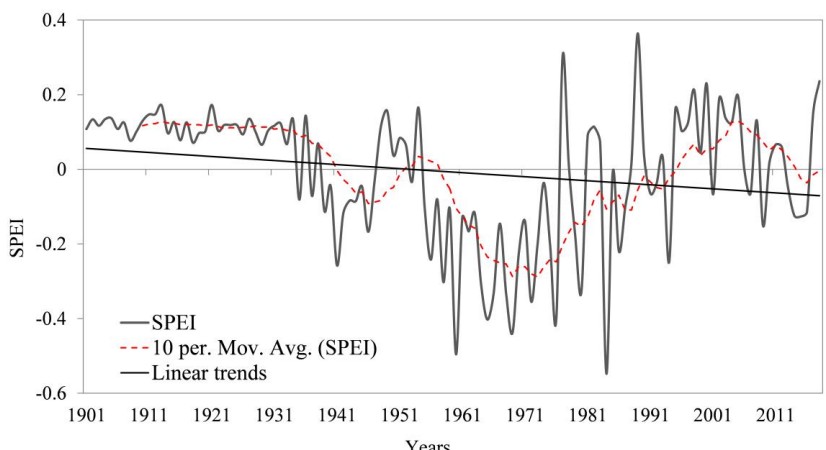

420            **Fig. 3 Trends of SPEI on the Qinghai-Tibetan Plateau from 1901 to 2018**

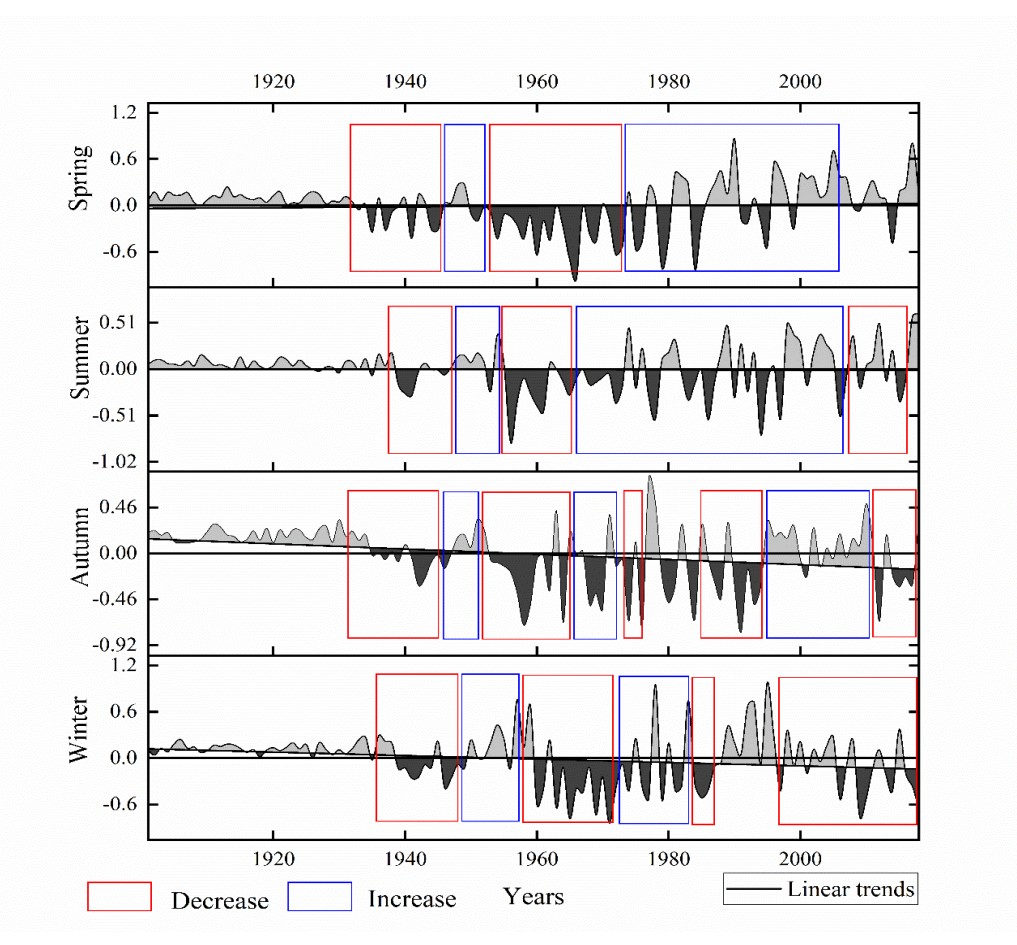

**Fig. 4 Seasonal trends of SPEI on the Qinghai-Tibetan Plateau during 1901-2018**

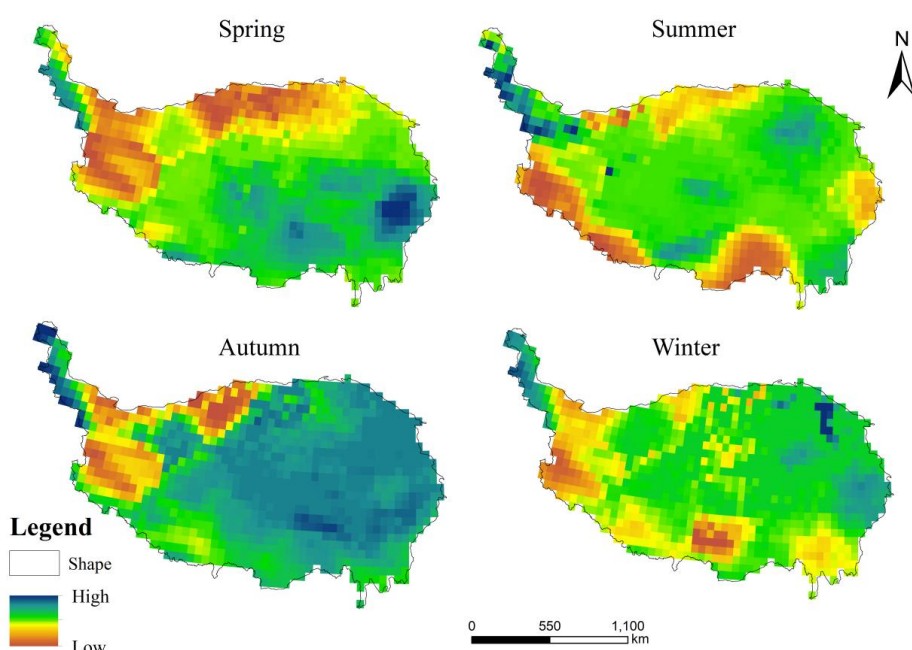

**Fig. 5 Regional trends in SPEI across seasons on the Qinghai-Tibetan Plateau**

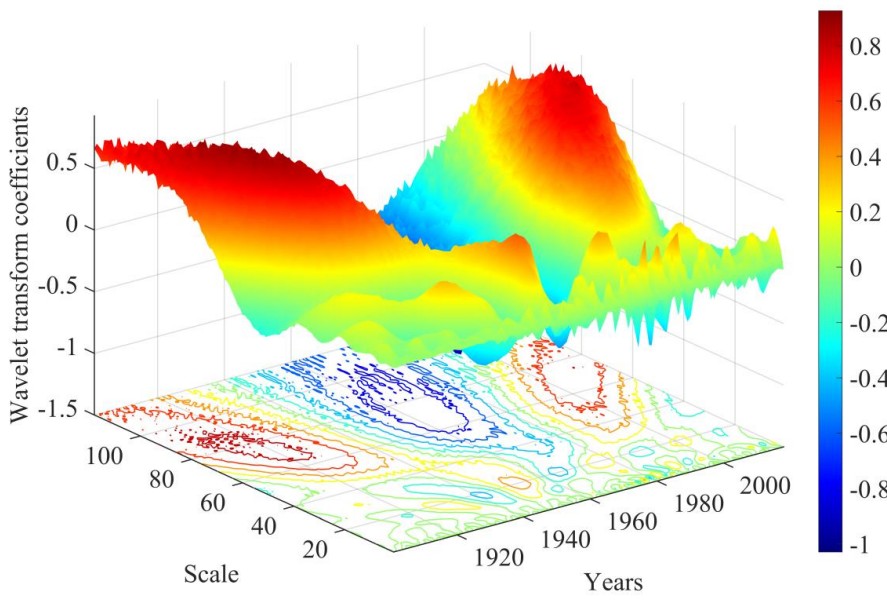


**Fig. 6 Stereogram of wavelet transform coefficients of SPEI for the Tibetan Plateau**


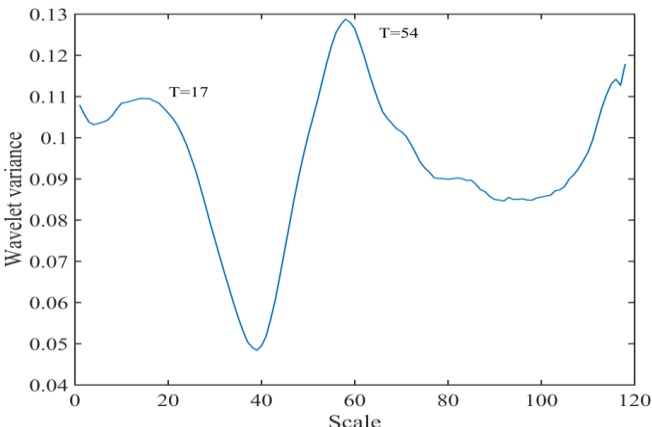

**Fig. 7 Stereogram of wavelet transform coefficients of SPEI for the Tibetan Plateau**

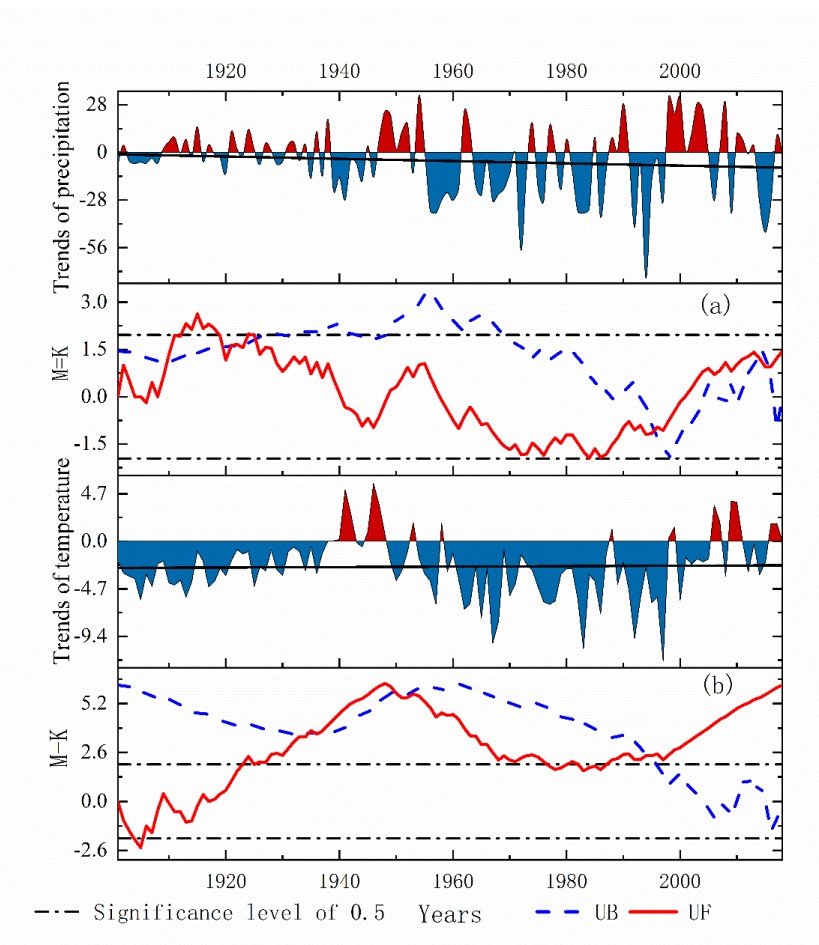

**Fig. 8 Trends of precipitation and temperature distance levels on the Qinghai-Tibetan Plateau from 1901-2018 and the results of the M-K test**





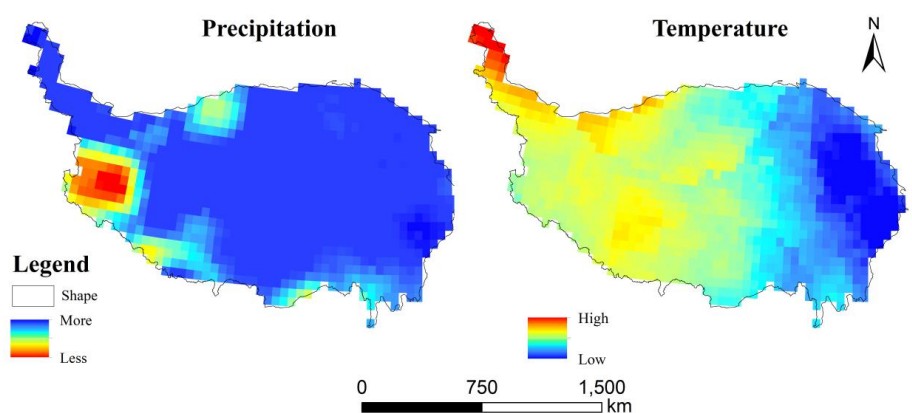

**Fig. 9 Regional variations in precipitation and temperature on the Qinghai-Tibet Plateau during 1901-2018**






**Table 1 Periods of SPEI change and their magnitude of change**

| SPEI trends | Year of beginning and ending | Range of changes/10a |
|---|---|---|
| **Decrease** | 1934-1946 | 0.17/a |
| | 1954-1972 | 0.13/a |
| | 2005-2016 | 0.18/a |
| **Increase** | 1946-1954 | 0.19/a |
| | 1972-2005 | 0.11/a |

**Table 2 Seasonal periods of changeof SPEI and their magnitude of change**

| Season | Trends | Year of beginning and ending | Range of changes /10a | Season | Trends | Year of beginning and ending | Range of changes /10a |
|---|---|---|---|---|---|---|---|
| **Spring** | Decrease | 1931-1944 | -0.11 | **Autumn** | Increase | 1946-1951 | 0.77 |
| | | 1952-1972 | -0.12 | | | 1964-1972 | 0.22 |
| | Increase | 1944-1952 | 0.25 | | | 1976-1985 | 0.17 |
| | | 1972-2005 | 0.20 | | | 1993-2010 | 0.17 |
| **Summer** | Decrease | 1938-1948 | -0.16 | **Winter** | Decrease | 1936-1949 | -0.27 |
| | | 1954-1965 | -0.09 | | | 1959-1971 | -0.53 |
| | | 2008-2015 | -0.36 | | | 1983-1987 | -1.80 |
| | Increase | 1948-1954 | 0.02 | | | 1998-2018 | -0.23 |
| | | 1965-2008 | 0.05 | | Increase | 1949-1959 | 0.52 |
| **Autumn** | Decrease | 1932-1946 | -0.24 | | | 1971-1983 | 0.55 |
| | | 1951-1964 | -0.32 | | | 1987-1998 | 0.15 |
| | | 1972-1976 | -1.05 | | | | |
| | | 1985-1993 | -0.73 | | | | |
| | | 2010-2017 | -0.23 | | | | |