# Peer review of "Drought evolution characteristics of the Qinghai-Tibet Plateau over the last 100 years based on SPEI"

_Natural Hazards and Earth System Sciences, 2021_

## Author Comment (AC2)

**Response Letter**

Dear Editor,

We appreciate your coordination of helpful and constructive reviews of our manuscript. We are confident that the referees' feedback and comments will help to eliminate imprecision and improve the manuscript. Below, we reply to their comments point by point. The reviewer comments are highlighted in black, while our responses are highlighted in blue. We numbered the reviewer comments to make it easier to refer to a response to another specific comment. R1C1, for example, stands for referee 1, comment 1. Figures and tables used in our response are included at the end of the document. We look forward to your editorial consideration of our response.

Best regards,

Shengzhen Wang

On behalf of all authors

**#1 Referee Comments**

*General comments*

The paper of Wang et al. addresses drought trends in the Tibetan Plateau using the SPEI index and attempts to provide some indications about the main temporal and spatial trends in the years 1901-2018. Nevertheless, the manuscript is very far from achieving its objectives, because it has many drawbacks leading me to suggest its rejection.

We thank the referee for the review and the constructive comments. We address here all the points raised in the review.

- The paper does not provide any novelty either concerning methodology or results. It mainly looks like a technical report, but rather confused.

- The paper is very poorly organized, with several serious drawbacks in every Section (please refer to specific comments).

- Methodology explanation has many unclear aspects. In particular, the spatial scale of the problem is not declared. If the authors use a spatially distributed database, I don't understand why they talk about spatial interpolation. Furthermore, please note that a

global SPEI database, based on CRU TS dataset v4.03, already exists:

https://spei.csic.es/database.html

- There's practically no discussion. The authors never try to explain the reason for the results they get. Furthermore, results are formally inaccurate (e.g., the term "significant" is used with excessive ease).

Therefore, I strongly suggest that the authors deeply rethink both research aims and the structure of their paper and try to take more advantage of the only original part of their study, consisting of retrieving drought hazard data, which could be more fruitfully utilized for validating global datasets.

Please find below my specific comments.

Abstract:

Thank you for your suggestion. Based on your suggestion, we have sorted out and improved the abstract section.

Introduction:

We used the data from the CRU dataset to calculate the SPEI index using the R code. We compared CRU4.01, CRU4.03 and other versions, and found that CRU4.03 is more applicable to the Tibetan plateau, so we used this version. We have included this comparative analysis process in the paper and further refined the introduction.

Methods:

Thank you very much for your suggestion. We have carefully described the tools and so on for the use of M-K in the article.

Section 2.1:

Thank you very much for your suggestion. We provide a detailed description of all the indices that appear

Section 2.2:

Thank you very much for your suggestion. We have added a description of the reasons for introducing the M-K method in the text.

Results:

Thank you very much for your suggestion.

1. We further illustrate the occurrence of terms such as "drought frequency" (Fig. 1 Fig. 2).

2. Additional analysis was conducted for the applicability of the drought index in the Tibetan Plateau (Fig. 2).

3. We provide a detailed discussion of the temporal and spatial trends of drought.

4.A study area overview map was added to illustrate the geographical location of the areas that appear in the text (L186).

Section4.4:

The analysis of periodicity is further introduced and discussed.

Discussion:

Thank you very much for your comments.

1.We have carefully combed through the discussion section.

2.A detailed description of the UFs appearing in the figure and the meaning of UB is given. (Fig.8)

**#1 Referee Comments**

***General comments***

I have reviewed the manuscript nhess-2021-73 "Drought evolution characteristics of the Qinghai-Tibet Plateau over the last 100 years based on SPEI" by Wang et al. The paper attempts at a description of drought features over the Tibet Plateau by means of a statistical analysis of the SPI index over a very long period of time. The topic of the research is up to date and coherent with the scope of the journal and the specific case study is potentially interesting but I regret to say that the current version of the manuscript represents for presentation, methodology, analysis and results commenting a poor contribution to the current and relevant scientific literature. I would not recommend the paper for publication and warmly invite authors for an in-depth revision of the research framework prior to a further submission.

We thank the referee for the review and the constructive comments. We address here all the points raised in the review.

1.Title: the title only refers to the SPEI index but in the end the paper also analyses the precipitation and temperature trends.

Thank you for your suggestion. We have given this careful consideration.

2)Abstract:it is a confuse description of the research idea and content of the paper, it highlights the fact authors need to revise the research framework.

Thank you for your suggestion. We have carefully revised the framework of the paper.

3)Introduction:   after a fair introduction of the current relevant literature, it confusedly presents the proposed research work.

Thank you for your suggestion. We have revised the introductory section.

4)Method:   the description of the methodology is not exhaustive. In the presentation of the results, I only saw representation of one single time series. What does it represent? Is it the average over the region? If so, I do not believe the average over a large region is sufficient to explain a behaviour. Provided the fact that in the results also gridded maps are represented, I guess more time series should be available, one for each cell of the grid. How do authors deal with the spatial dimension of the problem then? And also how do authors deal with the temporal dimension of the problem? It is well known the SPEI, such as SPI, can be assessed over different accumulation time scales, each of which has a specific significance. This specific issue is never discussed in the paper.

In the paper, we explored the trends of drought in time series using the average values for the whole study area of the Tibetan Plateau. We will add the analysis in spatial dimension, i.e., the trend of drought in each grid cell, in later improvements of the paper. In addition to this, we add a multi-scale characterization of drought.

4)Data and preprocessing: overall the data are very poorly described. The climate data are poorly presented, probably a graphical comparison between rain gauge and grid data could have been important to explain the need for the use of CRU 4.03. The geographical context is not described at all. It is not clear what drought hazard data are. Data processing: how? But more importantly why?

Thank you for your suggestion. We have improved the section on data and data pre-processing, especially the source of drought hazard data and its description. In addition to that the reasons for data preprocessing are explained.

5) Results and discussion: these sections are very poor for both the content and the interpretation provided. The feeling is that the authors are not familiar with statistical analysis of time series or at least with the interpretation of the relevant results. The time scale and the spatial scale of the problem are not considered properly, as previously mentioned. Figures are not well described, captions are not complete, frequently the reader does not know what is looking at and sometimes the interpretation of the results are wrong.

What is Figure 1 representing? The average SPEI? What is a typical annual drought frequency? How do we guess from Figure 1 and 2 there is a good agreement between SPEI and drought frequency (who is it assessed?) for the case study? Unless the legend of Figure 2 is wrong, the two panels (upper and lower) describe different spatial patterns. Drought frequency is large in the northern area which is instead the region with the lower SPEI (again what index are we looking at?).

Figure 1 represents the year-by-year trend of the average SPEI of the Tibetan Plateau. The drought frequency in Figure 1 represents the frequency of drought disasters recorded in that year. From Fig. 1, we can see that the SPEI values are smaller in the years with higher drought frequencies, so there is a good correspondence in the time series. In Figure 2, the years with more severe droughts in the disaster records are selected, and the number of droughts in the corresponding years is spatialized. Afterwards, this data was analyzed in comparison with the SPEI distribution of the corresponding years, and a good agreement was found. Based on this reason, we believe that SPEI has good applicability on the Tibetan Plateau.

Figure 4: we cannot assess temporal trends over short period of time (or at least we can but there is no meaning), so how authors assessed the interplay for an increasing and decreasing trend in the single seasonal time series? This makes no sense to my opinion.

Figure 4 divides the drought index from 1900 to 2020 into seasons, and further analyzes how the drought changes in the time series for each season. This is due to the fact that drought has a strong seasonality on the Tibetan Plateau, with different trends in each season.

Figure 5: the legend should illustrate number not just quality of the mapped attributes.

We have made changes to the legend in response to this comment. Thank you very much for your suggestion.

Figure 6-7: the wavelet analysis, which authors mentioned to engage in the abstract is very poorly undertaken. What is the physical meaning? How does it compare or add to the previous results of the time series analysis?

Thank you very much for your suggestion, and we have further supplemented the results of the wavelet analysis.

Figure 8: what does Figure 8 illustrate? According to the caption and axes titles, should be the trend of temperature and precipitation. But this should be an univocal number for each single time series. Additionally in the methodology authors never say how they assess the magnitude of the trend, which is not provided by the Mann-Kendall test. In the first and third panels, what does the pattern illustrate? Why are they depicted in blue and red?

The red area in the graph represents the average value of temperature (or precipitation) for the current year compared to the calendar year, while blue represents a low value. Our description of the picture was not clear and we have carefully revised it.

Thank you very much for your advice, it has helped me a lot in writing my paper. We hope that the revised manuscript has satisfactorily addressed all the concerns raised.